# Ultrafast transient infrared spectroscopy for probing trapping states in hybrid perovskite films

Ahmed M. El-Zohry[1,2✉], Bekir Turedi[3], Abdullah Alsalloum[3], Partha Maity[1], Osman M. Bakr[3], Boon S. Ooi[4] & Omar F. Mohammed[1✉]

Studying the charge dynamics of perovskite materials is a crucial step to understand the outstanding performance of these materials in various fields. Herein, we utilize transient absorption in the mid-infrared region, where solely electron signatures in the conduction bands are monitored without external contributions from other dynamical species. Within the measured range of 4000 nm to 6000 nm (2500–1666 cm$^{-1}$), the recombination and the trapping processes of the excited carriers could be easily monitored. Moreover, we reveal that within this spectral region the trapping process could be distinguished from recombination process, in which the iodide-based films show more tendencies to trap the excited electrons in comparison to the bromide-based derivatives. The trapping process was assigned due to the emission released in the mid-infrared region, while the traditional band-gap recombination process did not show such process. Various parameters have been tested such as film composition, excitation dependence and the probing wavelength. This study opens new frontiers for the transient mid-infrared absorption to assign the trapping process in perovskite films both qualitatively and quantitatively, along with the potential applications of perovskite films in the mid-IR region.

[1] Division of Physical Sciences and Engineering, King Abdullah University of Science and Technology (KAUST), Thuwal 23955-6900, Saudi Arabia. [2] Department of Physics, AlbaNova Center, Stockholm University, 10691 Stockholm, Sweden. [3] KAUST Catalysis Center, King Abdullah University of Science and Technology (KAUST), Thuwal 23955-6900, Saudi Arabia. [4] Photonics Laboratory, King Abdullah University of Science and Technology (KAUST), Thuwal 23955-6900, Saudi Arabia. ✉email: ahmed.elzohry@fysik.su.se; omar.abdelsaboor@kaust.edu.sa

Hybrid perovskite materials have recently attracted lots of attention due to their unique photo-physical properties and their high performances in various applications such as solar cells and light emitting diodes[1–17]. However, still controlling the amount of traps present in these materials especially upon making thin films is a challenging procedure[18,19]. The presence of trap states either through structural defects or other types can quench the charge carriers motilies inside the materials and thus reducing both the device's performance and its stability[8,18,20–23]. Various direct and indirect methods have been applied to track and quantify trap states such as electrical or optical measurements, however, most of these still have some drawbacks[8,18–22,24]. For instance, conductivity measurements have low time resolutions and can't distinguish between various carriers such as electrons and holes especially upon having close motilies[25]. Also, the most commonly used optical measurements such as time-resolved photoluminescence or transient absorption in the visible range couldn't afford direct spectral signatures for trapping states except providing variations of multi-exponential kinetic rates between different samples according to the estimated traps present in the investigated samples[1,2,7,13,15,26,27]. Thus, still there is a need for a direct transient optical method to track and quantify the trapping process in perovskite materials, and correlate their presence directly to charge dynamics. For that sake, we utilized mid-infrared (mid-IR) probe to monitor the charge dynamics of four different perovskite films with different compositions. Following the charge dynamics using femtosecond transient absorption (fs-TA) in the mid-IR has been used previously for several systems including metal complexes[28–31], organic dyes[32–37], metals[38], and semiconductors[21,25,39–43].

Basically for classical semiconductors, the electron's absorption in the conduction band with high density of states has a broad spectral signature extending from 3333 nm (3000 cm$^{-1}$) to 11,111 nm (900 cm$^{-1}$), in which other contributions from cationic or anionic molecular species present can be easily quantified[25,28,32,33]. The positive signature in the mid-IR upon photon excitation is ascribed to the presence of intra-transition of free electrons in/into the conduction band of the semiconductor used[25,43,44]. The transient mid-IR was used to follow trapped electrons at the mid-gap shallow states in the platinized $TiO_2$ system, in which IR emission is evolved in the IR region as a result of electron trapping process[25,43]. Recent mid-IR studies have been done on perovskite materials, however in those studies the authors focused on following the NH vibrational modes, present in the organic cationic part of the perovskites[20,21,26,45]. In contrast, in our selected mid-IR region, we don't have any contribution of the vibrational modes of the organic part, only transient signal of electrons in the conduction band, see Fig. S1.

Herein, we propose using fs-TA in the mid-IR region as a sensitive tool to follow the presence of traps in hybrid perovskite films. In the current study, various hybrid perovskite films have been synthesized and utilized to study the charge dynamics in the mid-IR region extending from ca. 4000–6000 nm (2500–1666 cm$^{-1}$). We found that perovskite films with methyl-ammonium cation (MA) and halides of Iodide (I) derivatives tend to emit mid-IR than other films with formamidinium (FA) and bromide (Br) derivatives, depending on the working conditions and the quality of the prepared films (see steady state measurements in Figs. S2–4). This study presents deeper understandings of charge dynamics in perovskite films, and potential applications of perovskite films in the mid-IR region.

## Results and discussion

Figure 1A shows the false 2-D plot of fs-TA in the mid-IR region with a central detection window of 5000 nm (2000 cm$^{-1}$) for $MAPbBr_3$ thin film using an excitation wavelength of 530 nm. At time zero, an intense positive signal appears due to the population of electrons in the conduction band as described for previous semiconductors such as $TiO_2$[28,32,43,46]. The contribution of holes in the valence band is expected to be minimum due the low energy of the probed light in the mid-IR range, however, further studies are needed to certify that. This positive signal decays exponentially toward zero within few nanoseconds. However, the extracted spectra show negative features at longer time scale >1.0 ns; see Fig. 1B. The extracted kinetic trace at 4900 nm shows a multi-exponential decay for the positive signal with an average lifetime of 120 ps, followed by a small negative feature beyond 2 ns; see Fig. 1C.

The same film ($MAPbBr_3$) was measured by fs-TA in the visible range, and an extracted kinetic trace at 550 nm corresponding to the ground state bleach (GSA) is compared with the extracted kinetic trace at 4900 nm from the mid-IR region; see Fig. 1D and Fig. S5. The comparison shows that both normalized kinetic traces from different spectral regions are very similar, except the presence of a new feature at the extracted kinetic trace from the mid-IR range; see Fig. 1C–D. The similarity between the two kinetic traces for $MAPbBr_3$ film confirms the validity of mid-IR signal to trace charge dynamics in perovskite films. However, the charge dynamics in the mid-IR region are not similar. For example, negative features at the red-part of the false 2D plot (at 4900 nm) in Fig. 1A, appears differently than in the blue-part (at 5120 nm). Extracting a kinetic trace at 5120 nm (1953 cm$^{-1}$) shows earlier conversion of positive to negative signals at ca. 200 ps; see Fig. 1C. Also, comparing this kinetic trace with the one extracted from the GSB in the visible region, shows different behavior than kinetic trace at 4900 nm; see Fig. 1D. This highlights the dependence of such negative feature on the probed spectral window.

Upon measuring the iodide-derivative, $MAPbI_3$ film, a detectable fs-TA mid-IR signal was also found, but with different behavior, see Fig. 2A, B. Interestingly, the transient mid-IR signal for $MAPbI_3$ film was changing over minutes time scale (minutes); see Fig. 2C. Thus, various kinetic traces were extracted at different times and compared together at the same probed wavelength. For instance, the fresh-irradiated film (~ 0 min.), appositive signal was measured until ca. 100 ps, and then a small negative signal started to emerge. The disappearance of the positive signal became faster with the longer the exposure process associated with an increase of the negative signal at early times; see Fig. 2C. For example, after 26 min of irradiation, the positive signal converted into a negative signal within 10 ps; see Fig. 2C. Interestingly, this process is reversible, in which switching off the irradiation for almost 8 min, and re-measure the dynamics again at the same irradiated spot (@ ~ 34 min in the Fig. 2C), the dynamics slowly started to be similar to the 10 min irradiation measurements; see Fig. 2C. In all cases, after the appearance of the negative signal, it decays later on to zero due to the expected recombination process, see Fig. 2C.

Upon measuring the same film in the visible range, a strong GSB signal at ca. 760 nm was observed overlapping with an ESA spectra extending from 650 to 850 nm, see Fig. S7. However, no unique change in dynamics has been observed in the visible range similar to the shown data in the mid-IR range. And upon comparing the extracted normalized visible kinetic trace at 760 nm with the ones from the mid-IR, it is evident that the visible kinetic trace is similar to the one from mid-IR at 26 min only at early times; see Fig. 2D. However, still at later times, the mid-IR signal switches its sign, but not the visible kinetic trace at 760 nm. The change of charge dynamics upon irradiation in the mid-IR region has been assigned in iodide-rich perovskite materials to halide-defects assisted by the low energy needed for defects

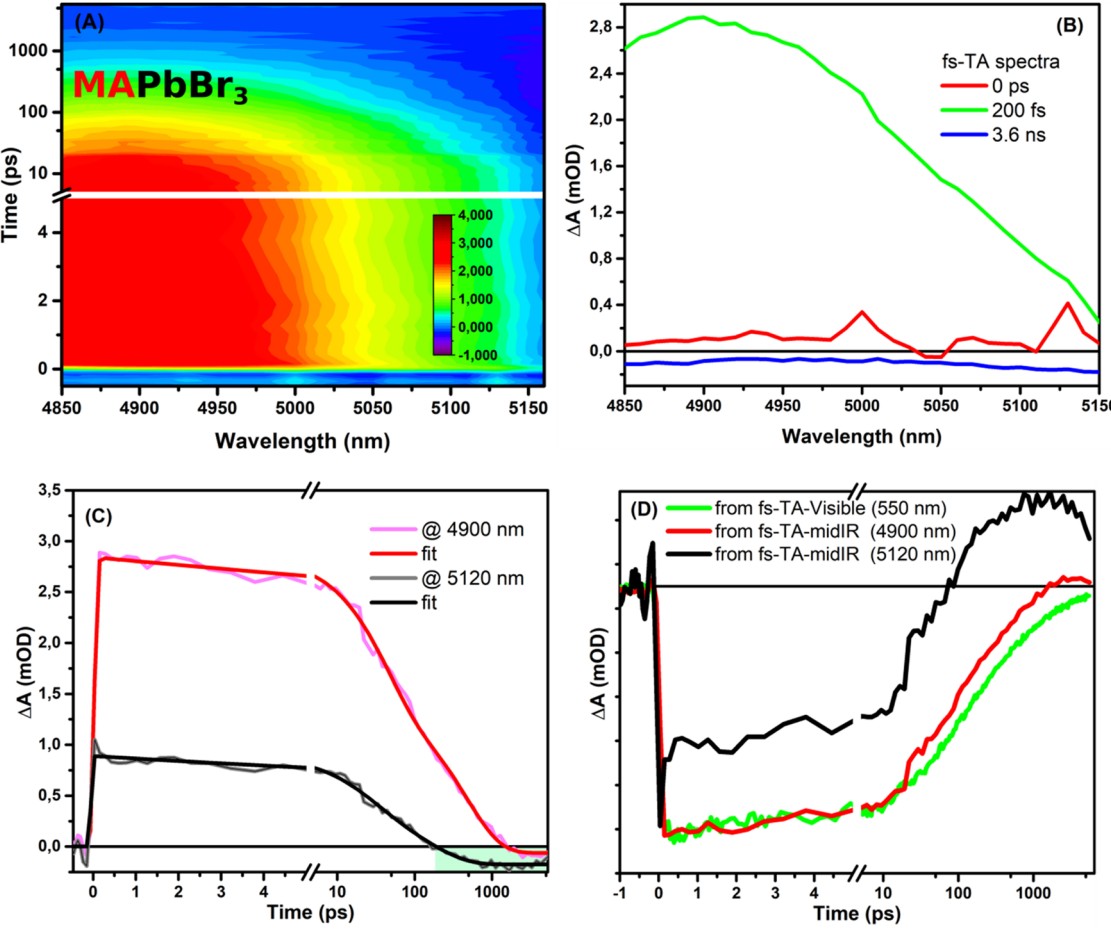

**Fig. 1 Transient mid-IR data for MAPbBr₃ film. (A)** 2D-false color plot of fs-transient absorption in the mid-infrared regions for MAPbBr₃ film using 530 nm as an excitation source. **B** Extracted spectra for transient spectra in mid-infrared region, the spectra are corrected for the wavelength scale. **C** Extracted kinetic trace at 4900 and 5120 nm with corresponding fitting, green shaded area highlights the appearance of negative signal. **D** Comparison between normalized kinetic traces in the infrared and visible ranges for MAPbBr₃ film.

formation[18,47–49]. This comparison highlights also that this appearance of negative features in the mid-IR region is a dynamical process and also can be reversible (MAPbI₃ case).

The extracted kinetic trace at 4900 nm from the MAPbI₃ shows stronger and earlier negative signal than the one shown in MAPbBr₃ case. Moreover, for further confirmation of the working conditions, a reference silicon wafer substrate was measured under the same procedure, and no negative features have been detected, see Fig. S8. Thus, we assign these strong negative signatures in the mid-IR for MAPbI₃ film to the emissive trapping process near the VB of perovskites, confirming previous studies about tendencies of iodide-based perovskites to form more trap states than Br ones[1,8,15,18].

To scrutinize our interpretation about the sensitivity of mid-IR toward trapping process, we performed the same measurements on other perovskite films including FAPbBr₃, FAPbI₃ and mixture of their halides. For the FAPbBr₃ film, the mid-IR signal shows primarily a strong negative signal close to time zero converting into a positive signal with a lifetime of ca. 60 fs, which has been assigned to the exciton thermalization/dissociation process;[7] see Fig. 3A, B. Previously, in the MAPbBr₃ film, exciton binding energy seems to be smaller, thus, no detection of exciton dissociation process could be seen; see Fig. 1. The charge recombination in the FAPbBr₃ film (decay of the TA signal) has been fitted with multi exponential behavior, giving an average lifetime is about 10 ps, see Fig. 3B.

For the FAPbI₃ film, similar observation was estimated for the lifetime needed for exciton dissociation in FAPbI₃ film; see Fig. 3C, D.

However, instead of charge recombination, the measured positive signal converted again to a negative signal due to a trapping process of time component ca. 8.5 ps; see Fig. 3D. Then the trapped electrons recombine slowly with a lifetime higher than 1 ns. It is clear now that this negative signature in the mid-IR range is associated with the iodide derivative of hybrid perovskite films.

To verify the role of iodide anion for the formation of emissive trapping centers, we also synthesized other set of various perovskite films of different ratios between the iodide and Br halides (MAPbIₙBr₃₋ₙ) to investigate the effect of doping with iodide ions on the presence of such negative signal. Figure 4A shows the kinetic traces of mid-IR signals for four films of MAPbIₙBr₃₋ₙ, in which n varies from 0 to 3. The extracted kinetics at 4900 nm show the appearance of negative signals at different times ranging from 100 ps to 1 ns depending on the amount of iodide halide present, in which higher content of iodide shows faster appearance of negative mid-IR signal, showing that the iodide content in perovskite films controls the appearance of the negative signal (emission of mid-IR).

To study the dependence of appearance of these negative signals on the energy of the mid-IR probe, the MAPbI₃ film is excited at 520 nm (300 μW), and probed at various mid-IR energy ranging from 4000 nm to 6000 nm, as shown in Fig. 4B. As expected, the appearance time of the negative signal depends on the utilized mid-IR energy, in which the switching points from positive to negative signal happen at ca. 500 ps when using 4000 nm, and at ca. 20 ps upon using 6000 nm probes.

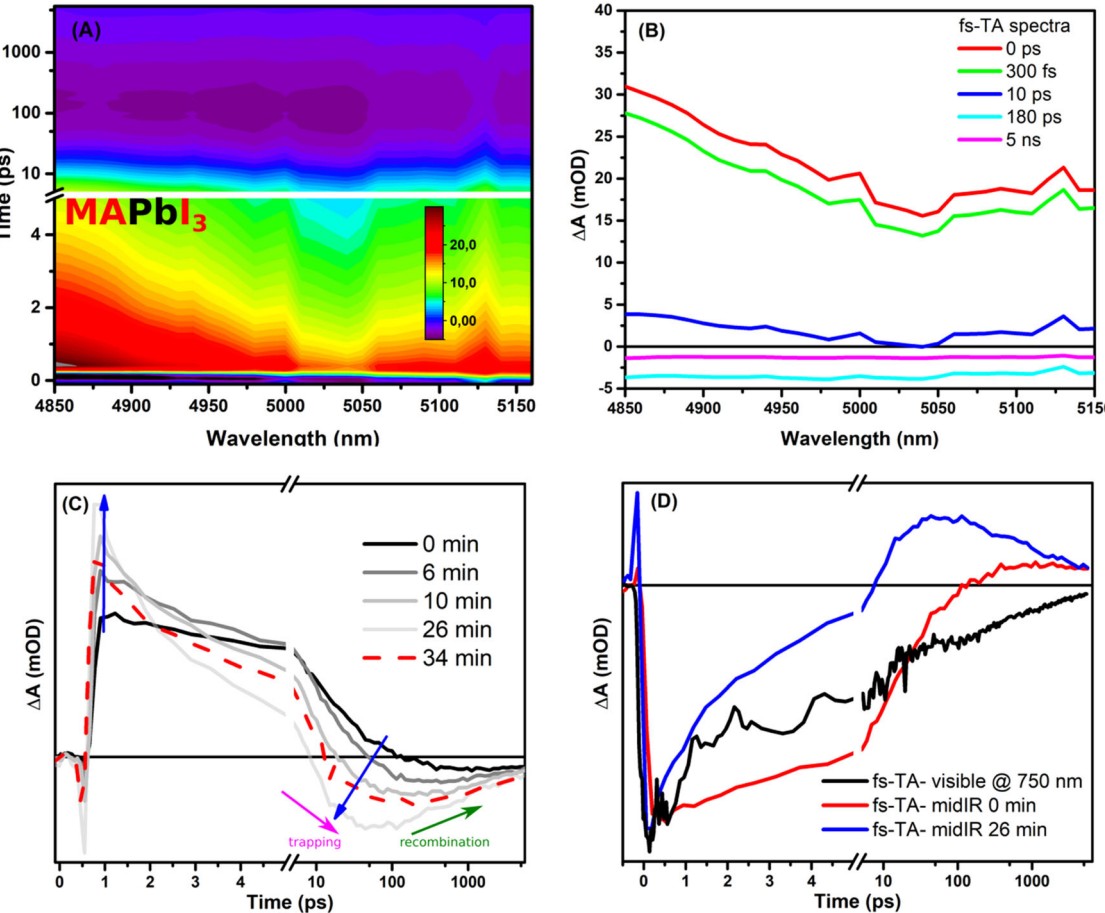

**Fig. 2 Transient mid-IR data for MAPbI$_3$ film.** (**A**) 2D-false color plot for fs-transient absorption in the mid-infrared regions for MAPbI$_3$ film using 530 nm as an excitation source after 26 minutes of irradiation. **B** Extracted spectra for transient spectra in mid-infrared region (**A**), the spectra are corrected for the wavelength scale. **C** Extracted kinetic trace at 4900 nm at different irradiation time shown in minutes for MAPbI$_3$ film, using 520 nm with excitation power of 500 μW. **D** Comparison between normalized kinetic traces in the infrared and visible ranges for MAPbI$_3$ film.

Furthermore, by changing the excitation wavelength from 410 nm to 520 nm for another MAPbI$_3$ film, the extracted kinetic traces at 4500 nm show a dependence of the kinetic decay with the wavelength used; see Fig. 4C. The appearance of negative signal is faster upon using 410 nm then became slower with 440 nm, and 520 nm respectively. Upon using the excitation light at 520 nm, the kinetic trace decay to zero with no signature of negative signal, despite that higher power was used, almost 20 times higher (300 μW) than at 410 nm; see Fig. 4C. This excitation energy dependence shows that the higher the electron can be promoted in the excited state, the higher chances to be trapped in emissive centers. Apparently, these emissive centers are formed during the excitation process of perovskite films.

From the above mid-IR transient measurements, the following mechanism can be drawn, see Fig. 5. Upon exciting the perovskite film, electrons in the CB should absorb the following mid-IR probe to populate various vibrational levels in the excited state, giving a positive transient absorption signal. Due to the presence of trap states within the bandgap of the perovskite film, electrons in the CB decay non-radiatively to the ground state (channel 1 in Fig. 5), in which the transient signal decays to zero as shown in FAPbBr$_3$, Fig. 3. Interestingly, upon synthesizing perovskite films using chemical species such as MA$^+$ or I$^-$ or both of them as in MAPbI$_3$, negative transient signal in the mid-IR region starts to appear. And since perovskite films have no characteristic features in this mid-IR region, this negative transient signal can be only due to emission of mid-IR signal or mid-IR gain (channel 2 in

Fig. 5). It has been already established in the literature that MA/iodide perovskite films show more potential to form trap states than other derivatives (FA/Br)[1,15]. In addition, these mid-IR gain can be controlled by the incident excitation energy, see Fig. 4C. Thus, we postulate that in MAPbI$_3$ films, different kind of emissive trap states are additionally formed related to ion migration and formation of transient phonon modes that are not present for instance in the FAPbBr$_3$ films[1,15,50].

Interestingly, to detect these mid-IR emissive states, suitable probe energy should be utilized as shown in Fig. 4B. The probe energy data presents the influence of the energy carried by the probing photons to free/decouple the trapped electrons if sufficient energy is present. For instance, the probed pulse at 4000 nm (2500 cm$^{-1}$) can liberate the trapped electron more efficient than at 6000 nm (1666 cm$^{-1}$), and the appearance of mid-IR negative signal upon using 4000 nm (2500 cm$^{-1}$) will not appear early, until 500 ps. In the same way, the appearance of negative signal at 6000 nm (1666 cm$^{-1}$) is much faster, due to the low energy carried by the probe pulse to liberate the trapped electron in the emissive states, instead allowing for deactivation through the IR stimulated emission. These observations are consistent with a previously proposed mechanism that trapped electrons can be excited thermally if the energy difference between the trap state and the CB is small, <50 meV[25]. This also illustrates the incapability of transient absorption in the visible region to detect such a trapping process due to the higher energy carried by the visible probed light, that have the potential to liberate the trapped

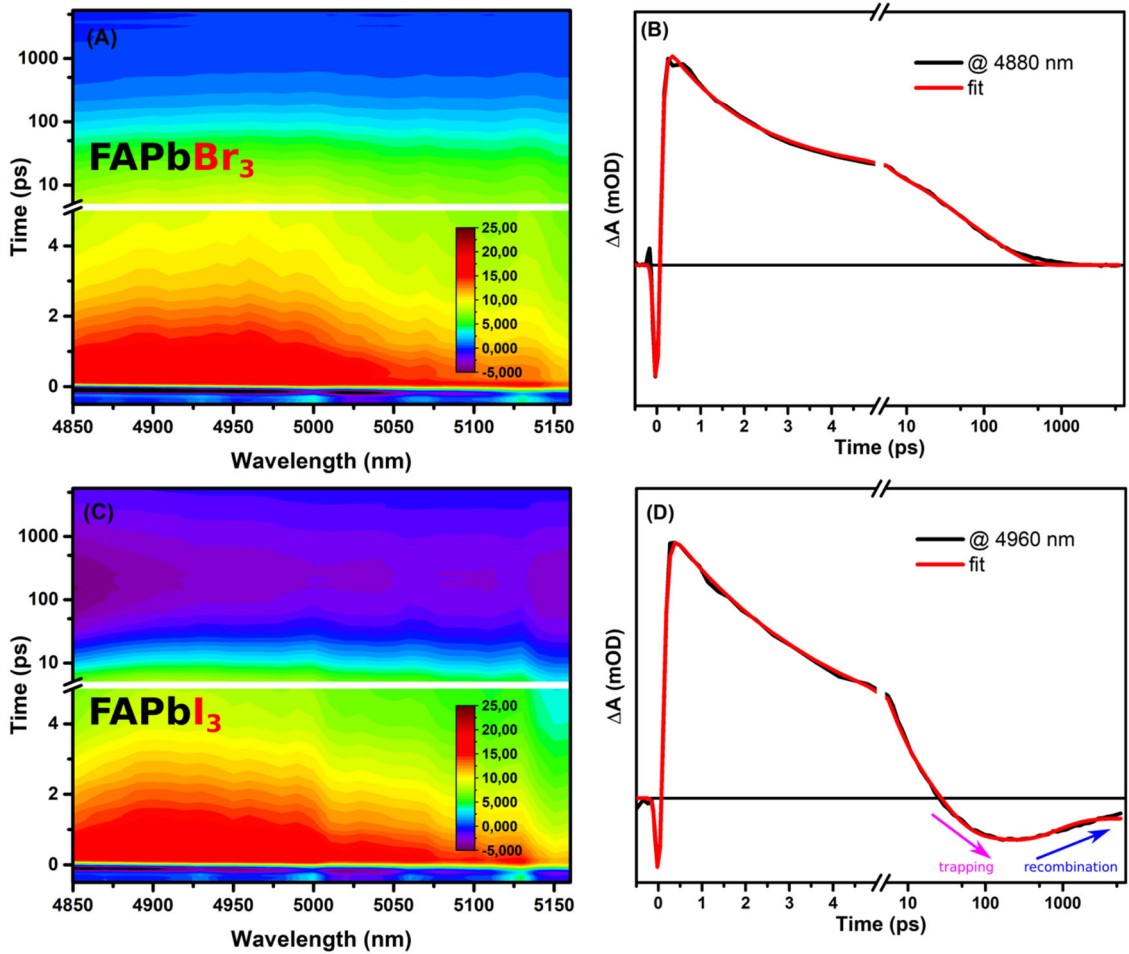

**Fig. 3 Transient mid-IR data for MAPb(Br₃)/I₃ films.** (**A**) 2D-false color plot for fs-transient absorption in the mid-infrared regions for FAPbBr₃ film using 530 nm as an excitation source. **B** Extracted kinetic trace at 4880 nm for FAPbBr₃ film. **C** 2D-false color plot for fs-transient absorption in the mid-infrared regions for FAPbI₃ film using 530 nm as an excitation source. **D** Extracted kinetic trace at 4960 nm for FAPbI₃ film.

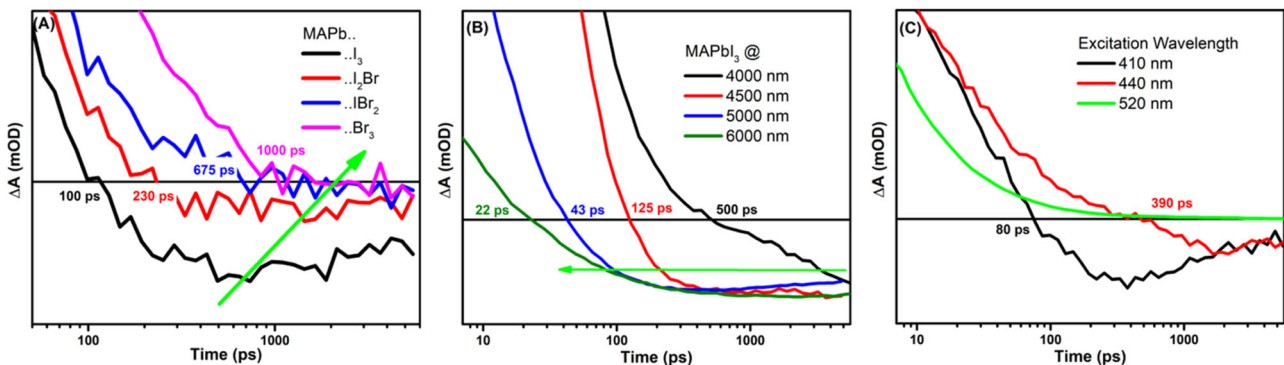

**Fig. 4 Dependence of various factors on the observed transient mid-IR signal.** Dependence on chemical composition: (**A**) Normalized kinetic traces at 4900 nm extracted form MAPbI_nBr_{3-n} films using 520 nm as excitation showing the appearance of negative signals. Dependence on mid-IR detection window: (**B**) Normalized Kinetic traces for MAPbI₃ film under excitation of 520 nm and at different probing wavelengths showing the appearance of negative signals. Dependence on excitation wavelengths: (**C**) Normalized kinetic traces for MAPbI₃ film under various excitation wavelengths showing the appearance of negative signals.

carriers into higher excited state, producing undistinguishable signal for the trapping process in the visible region, in which the excited electrons can be only deactivated via non-emissive trapping centers. According to the current range of probed energy utilized, it is expected that the high energy probe >1 eV will lead to deactivation through non-radiative centers, while <0.3 eV will stimulate mid-IR emission, see Fig. 5.

Moreover, upon using higher band-gap excitations such as 410 nm, the probability of electron trapping in these emissive states is increased despite the excitation intensity used, matching with the expected distribution for the states of trap-density present. We also show that continuous irradiation at high excitation energy for the MAPbI₃ film (Fig. 2C) increases the rate of trapping (channel 2 in Fig. 5), as well as the intensity of the transient

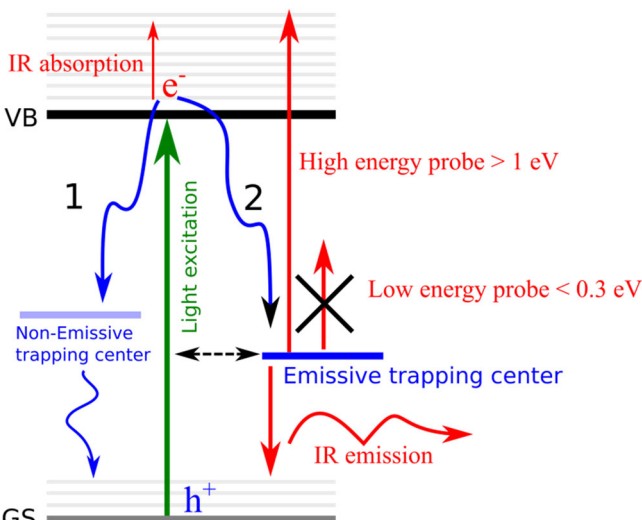

**Fig. 5 The overall scheme.** Schematic representation for the mechanism of charge recombination and charge trapping in various perovskite films especially (MAPbI₃). See text for more information.

signal. This indicates toward the exciting correlation between the light irradiation and ion migration process in perovskite films[51]. This means that changes in the perovskite lattice by the incident light (depending on energy) can lead to the formation of these emissive trapping states at different energy levels above the VB.

## Conclusion
We show herein for the first time that transient absorption in the mid-IR region is a suitable spectroscopic tool to explore trapping process in addition to follow the behavior of free electrons in the conduction band without other contributions of other species (reduced/oxidized species). Significantly, the selected region of the mid-IR spectrum can be utilized to follow the actual trapping process of electrons by detecting the negative appearance of the mid-IR transient signal, which is likely due to the formation of emissive trap states. We also figure out that similar measurements in the visible region could not be monitored due the energy of the probed light that can liberate the trapped carriers into higher excitation levels, providing additional complexity to distinct between tarped carriers and other species such as excitons and free carriers absorption. Interestingly, these emissive trap states in the mid-IR can be controlled by film quality, film chemical composition, and utilized bandgap excitation energy. This work will open frontiers toward understanding and controlling the nature of trapping centers in perovskite films, along with the potential of iodide-based perovskite films to generate emission in the mid-IR range.

## Methods
**Film preparations**. Preparing the perovskite films: the CaF₂ substrates were cleaned with DI water, acetone, and IPA, followed by a 10 min UV ozone treatment. FAPbBr₃ and FAPbI₃ thin films were prepared using a modified antisolvent dripping technique[52]. 1.1 M of FAX and PbX₂ were dissolved in DMF:DMSO (9:1 ratio), and 100 μl of the solution was spin-coated for 15 s at 4000 rpms. 300 μl of toluene was drop-casted during the sixth second, and the films were annealed immediately after the spin-coating process for 10 min at 170° for FAPbI₃ and 100° for FAPbBr₃.

**Femtosecond transient absorption setup**. Briefly, an excitation wavelength of 410–520 nm was utilized, while the probe light was also changing from 4000 nm to 6000 nm. The power used for the excitation wavelength depends on the utilized wavelength but was changing from ca. 100–300 μW[28,37,53–55]. The mid-IR light was detected on a N₂-cooled CCD that is sensitive to mid-IR photons.

## Data availability
All relevant data are available from the corresponding authors upon request (A.M.El-Z. & O.F.M.).

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

## Acknowledgements

The authors thank for the financial support provided by KASUT to carry out this work.

## Author contributions

A.M.El-Z. designed the concept of the paper, measured the mid-IR data wrote the paper, and put the overall scheme. B.T. and A.A. prepared the perovskite films and revised the paper. P.M. performed the transient data in the visible range along with revising the paper. O.M.B. supervised the project and revised the paper. B.S.O. supervised the project and revised the paper. O.F.M. supervised the project and revised the paper.

## Funding

## Competing interests

The authors declare no competing interests.
