## [Peer Review File · Communications Chemistry]

Reviewers' comments:

Reviewer #1 (Remarks to the Author):

El-Zhory et al., utilized transient absorption in the mid-infrared region to monitor recombination and trapping processes of the excited carriers. This report further reveals a clear distinction between recombination and trapping process. By studying various perovskite compositions, they found that iodide-based perovskite shows more tendencies to trap the excited electrons than the perovskite materials containing bromide. This study is important to the community as it opens up the possibilities to assign the trapping process in the perovskite film and therefore the manuscript can be accepted after the minor revision.

1. It is important to mention what energy the mid-IR wavelengths corresponds to. This will make it easier to relate to scheme 1.
2. There are some typo errors in the manuscript. For instance, "tarped" instead of "trapped". So, please correct them.
3. Authors are recommended to cite Chem. Rev. 2019, 119, 5, 3036–3103 and Adv. Energy Mater., 2021, 11, 2003489.

Reviewer #2 (Remarks to the Author):

The manuscript from El-Zohry et al on mid-infrared transient absorption spectroscopy of perovskite films conveys the presence of a negative ΔA signal in the transient response of these materials depending on the pump photon energy and the composition of the materials (particularly associated with iodide). On the conclusions based upon the data, the manuscript is reasonable, although I do think there are parts which should be clearer and the conclusions should be linked more directly to relevance of devices. I would suggest publication after minor revisions.

- 1) It is not so clear and obvious to me that the mid-IR signal should represent just the electrons. The effective masses and mobilities of the electrons and holes are similar, so why should the response be isolated to electrons. This is not very clear in the text.
- 2) is the negative MIR signal stimulated emission? Another way to state this is that the the sample show MIR optical gain. Can this emission actually be detected when not stimulated by a probe?
- 3) are there problems with the way that the iodide material is synthesized which give rise to trap states? What is the microscopic picture of the material defect?
- 4) several times the idea that a suitable probe energy for trap states should be used is emphasized. I would deemphasize this as it is clear that spectral information can only be had with a resonant probe; kinetic information may be obtained otherwise.

Point-by-point reply to reviewers

Please note that our reply will be in blue color

Reviewer #1 (Remarks to the Author):

El-Zhory et al., utilized transient absorption in the mid-infrared region to monitor recombination and trapping processes of the excited carriers. This report further reveals a clear distinction between recombination and trapping process. By studying various perovskite compositions, they found that iodide-based perovskite shows more tendencies to trap the excited electrons than the perovskite materials containing bromide. This study is important to the community as it opens up the possibilities to assign the trapping process in the perovskite film and therefore the manuscript can be accepted after the minor revision.

Many thanks for the reviewer appreciation and hopefully this work will be beneficial for the perovskite community.

1. It is important to mention what energy the mid-IR wavelengths corresponds to. This will make it easier to relate to scheme 1.

Thanks for the reviewer comment, we added a mid-IR energy scale according to our measurements upon discussing the proposed scheme.

2. There are some typo errors in the manuscript. For instance, “tarped” instead of “trapped”. So, please correct them.

Yes, this will be done along with correcting other words.

3. Authors are recommended to cite Chem. Rev. 2019, 119, 5, 3036–3103 and Adv. Energy Mater., 2021, 11, 2003489.

Thanks for the reviewer recommendation; these valuable references have been added.

Reviewer #2 (Remarks to the Author):

The manuscript from El-Zohry et al on mid-infrared transient absorption spectroscopy of perovskite films conveys the presence of a negative ΔA signal in the transient response of these materials depending on the pump photon energy and the composition of the materials (particularly associated with iodide). On the conclusions based upon the data, the manuscript is reasonable, although I do think there are parts which should be clearer and the conclusions should be linked more directly to relevance of devices. I would suggest publication after minor revisions.

Many thanks for the reviewer recommendation and hopefully after the revision, this work will be beneficial for the perovskite community.

1) It is not so clear and obvious to me that the mid-IR signal should represent just the electrons. The effective masses and mobilities of the electrons and holes are similar, so why should the response be isolated to electrons. This is not very clear in the text.

Thanks for the valuable comment. Yes, we do agree completely with the reviewer point. However, this mid-IR spectral sensitivity towards electrons rather than holes has been extensively studied in traditional semiconductors such as TiO₂ (Check Refs. 22, 26, 30/31, and 41-43 in the manuscript). In these studies the hole absorption has been limited to near IR window, while the free electrons can have spectral features until mid-IR region. Moreover, the absence of any spectral IR features in the ground state support the sensitivity of the mid-IR towards free electrons rather than holes.

2) is the negative MIR signal stimulated emission? Another way to state this is that the the sample show MIR optical gain. Can this emission actually be detected when not stimulated by a probe?

Thanks for the reviewer comment. The mid/IR optical gain term can be utilized and has been added as an alternative, please check the manuscript. We do believe an IR emission can be detected, however due to the short lived species of these states, the detection of these IR on normal detectors will be hard, thus, further studies are needed to develop such potential applications of perovskite films in the mid-IR range. A short text has been added to highlight these points in the text.

3) are there problems with the way that the iodide material is synthesized which give rise to trap states? What is the microscopic picture of the material defect?

Thanks for the comment. The iodide films are fabricated by following the same recipe with a previous paper (<https://www.sciencedirect.com/science/article/pii/S254243511930217X>) and for the FAPbI₃ films the similar approach has been introduced. We did not use any kind of passivation for the films with iodide or bromide to detect the inherent properties of them without additional effects which might screen them. The problem is not with the synthesizing the iodide perovskite films, rather than the inherent properties of iodide ions towards ion migration within the perovskite lattice upon irradiation forming trap states within the bandgap of perovskites. Such understandings have been concluded from several previous studies about the chemical structure of perovskite films. We have cited these papers in our manuscript for better illustration about our proposed scheme.

4) several times the idea that a suitable probe energy for trap states should be used is emphasized. I would deemphasize this as it is clear that spectral information can only be had with a resonant probe; kinetic information may be obtained otherwise.

Thanks for the reviewer comment, we modified the text in the manuscript accordingly.

REVIEWERS' COMMENTS:

Reviewer #2 (Remarks to the Author):

Questions raised in review have mostly been answered. I remain unconvinced by the assignment of the electron-exclusive signature (earlier work primarily focused on materials with very different m^*), but the remaining data is useful on its merits.